# Urinary Antibiotics and Dietary Determinants in Adults in Xinjiang, West China

**DOI:** 10.3390/nu14224748

**Published:** 2022-11-10

**Authors:** Lei Chu, Hexing Wang, Deqi Su, Huanwen Zhang, Bahegu Yimingniyazi, Dilihumaer Aili, Tao Luo, Zewen Zhang, Jianghong Dai, Qingwu Jiang

**Affiliations:** 1School of Public Health, Xinjiang Medical University, 567 Shangde North Road, Urumqi 830000, China; 2Key Laboratory of Public Health Safety of Ministry of Education, School of Public Health, Fudan University, Shanghai 200032, China

**Keywords:** antibiotics, adults, food, water, determinants

## Abstract

The Xinjiang autonomous region, located in west China, has a unique ethnic structure and a well-developed livestock industry. People in this region have a high risk of exposure to antibiotics, but the exposure level to antibiotics in relation to dietary determinants is unknown. In this study, 18 antibiotics, including four human antibiotics (HAs), four veterinary antibiotics (VAs), and 10 preferred veterinary antibiotics (PVAs) were detected in the urine of approximately half of the 873 adults in Xinjiang, including Han Chinese (24.6%), Hui (25.1%), Uighur (24.6%), and Kazakh (25.7%). Logistic regression was used to analyze the association between antibiotic exposure levels and adult diet and water intake. The detection percentage of antibiotics in the urine of adults in Xinjiang ranged from 0.1% to 30.1%, with a total detection percentage of all antibiotics of 49.8%. HAs, VAs and PVAs were detected in 12.3%, 10.3%, and 40.5%, respectively. Fluoroquinolones were the antibiotics with the highest detection percentage (30.1%) and tetracyclines were the antibiotics with the highest detected concentration (17 ng/mL). Adults who regularly ate pork, consumed fruit daily, and did not prefer a plant-based diet were associated with thiamphenicol, norfloxacin, and fluoroquinolones, respectively. These results indicated that adults in the Xinjiang autonomous region were extensively exposed to multiple antibiotics, and some types of food were potential sources of exposure. Special attention should be paid to the health effects of antibiotic exposure in humans in the future.

## 1. Introduction

Spatial modeling estimates that cumulative global consumption of antibiotics increased by 46% from 2000 to 2018 [1]. China accounts for a high percentage of global antibiotic use [2]. In addition to clinical use, the vast majority of antibiotics are used to treat and prevent bacterial infections in animal husbandry and aquaculture. It is expected that by 2030, as much as 105,596 tons of antibiotics will be used in animal food production [3,4]. In addition, studies have shown that antibiotics are commonly detected in foods of animal origin (e.g., edible meat, milk, aquatic products) at a rate of 39.2%, and are also detected in the aquatic environment [5]. As a result, antibiotics may re-enter the human body through unintentional ingestion of antibiotic-contaminated food or drinking water, resulting in long-term persistent low-dose antibiotic exposure, with potential adverse effects on human health.

The potential adverse effects on human health of continuous exposure to low doses of antibiotics in the environment have begun to attract the attention of scholars in the research community. A number of studies has investigated antibiotic exposure in humans by analyzing antibiotics in biological samples, and extensive antibiotic exposure was reported [6,7,8]. Xinjiang is located in northwest China, and has a unique ethnic structure and a well-developed livestock industry; thus, there is a higher preference for livestock and poultry meat in this region. As a result, residents of the Xinjiang region may be at greater risk of antibiotic-containing foods of animal origin. However, to date, antibiotic exposure in Xinjiang residents remains unclear.

Exposure to antibiotics can occur through food consumption and may be associated with sociodemographic factors. It is unclear which factors are most associated with antibiotic concentrations. We sought to investigate the antibiotic exposure profile in adults in Xinjiang and to explore the potential determinants of urinary antibiotics in adults using biomonitoring data of urinary antibiotics.

## 2. Materials and Methods

### 2.1. Study Population

Since 2018, our group has conducted the National Key Research and Development Program “The Xinjiang Multiethnic Cohort (XMC) study” in Xinjiang [9]. A sub-sample was recruited from the Yili region during the baseline survey in 2019. The recruited individuals were the residents of three townships in Huocheng County of Ili (including Langan Township, Sarbulak Township, and Luchaogou Township), who were randomly recruited according to the local main ethnic groups. Inclusion criteria were as follows: adults aged 35–75 years; residents who had lived locally for three years or more; not being in an acute state of illness; and not having used antibiotics within one month (to minimize the effects of clinically used antibiotics). After excluding participants with severe liver or kidney disease or mental illness, and those with incomplete questionnaires or whose urine or blood samples were not collected, a total of 873 adults with complete data were included in this study. The subjects completed a questionnaire, provided a 12 mL urine sample, and underwent a physical examination. All participants signed an informed consent form and were approved by the Ethics Committee of the Xinjiang Uygur Autonomous Region Hospital of Traditional Chinese Medicine (2018XE0108).

### 2.2. Questionnaire and Medical Information Collection

The baseline questionnaire of the Northwest Natural Population Cohort Study was used to investigate the demographic characteristics (gender, age, education level, per capita monthly expenditure, etc.), dietary status (source of drinking water, eating eggs, drinking milk and goat’s milk, eating pork, eating beef, eating mutton, eating seafood, eating vegetables, eating fruits, etc.), and lifestyle (smoking, drinking alcohol, physical activity, etc.). All questionnaires were completed by formally trained enumerators.

Medical information included body mass index (BMI), hospitalization information, and antibiotic use within one month. Body measurements were obtained according to standard anthropometric measurements, with height and weight readings accurate to 0.1 cm and 0.1 kg, respectively; waist circumference was taken at the midpoint from the upper edge of the hip bone to the lower edge of the rib cage on both sides, and measured during calm exhalation, with a reading accurate to 0.1 cm; hip circumference was measured as the circumference of the widest part of the hip on the horizontal plane, with a reading accurate to 0.1 cm. Body fat percentage and muscle mass were measured by bioelectrical impedance (instrument model: TANITA DC-430MA, Tokyo, Japan). Blood pressure was measured at rest using an electronic sphygmomanometer (model: Omron HEM-7133) at the brachial artery of the right upper arm of the study subjects.

### 2.3. Urine Collection and Analysis

Eighteen antibiotics from five major classes commonly detected in previous studies due to possible human exposure were measured [10,11]. These included four tetracyclines, four fluoroquinolones, three macrolides, four sulfonamides, and three chloramphenicols; four veterinary antibiotics (VAs), four human antibiotics (HAs), and ten preferred veterinary antibiotics (PVAs).

Morning urine (12 mL) was collected from the study participants and the samples were stored on site in a −40 °C refrigerator in the dark and frozen after collection. Prior to analysis, 1 mL of the urine sample was taken and sent to the laboratory for analysis. The concentrations of the 18 antibiotics in urine were determined using ultra-performance liquid chromatography coupled with high-resolution quadrupole time-of-flight mass spectrometry (UPLC-Q/TOF MS) [12]. Twenty microliters of isotopic internal standard was added to 1.0 mL of urine and hydrolyzed with β-glucuronidase, then purified by an OASIS HLB 96 solid phase extraction plate and analyzed by the UPLC-Q/TOF MS method. All antibiotics were separated on an HSS T3 column. The three chloramphenicol antibiotics were separated in negative ionization mode using acetonitrile-water as the mobile phase and in positive ionization mode using methanol-water as the mobile phase with 0.1% formic acid. Ninety-six samples were analyzed in each batch using 96-well solid phase extraction plates, including 92 real urine samples, 2 solvent blank samples, and 2 10 ng/mL spiked urine samples. Solvent blanks and spiked urine samples were analyzed together with the real urine samples. The solvent blanks were used to monitor background interference and the spiked urine samples were used to monitor precision and accuracy. The limits of detection (LOD) and limits of quantification (LOQ) were defined as signal-to-noise ratios of 3 and 10, respectively. The LOD and LOQ for all antibiotics ranged from 0.04 to 1.31 ng/mL and 0.13 to 4.37 ng/mL, respectively. No background interference from any of the antibiotics was observed. The spiked recoveries of the 18 antibiotics in urine ranged from 78.2% to 117.3%, with relative standard deviations of 8.2% to 15.7%.

### 2.4. Dietary Characteristics

Xinjiang is a multi-ethnic region, with Han Chinese, Uyghurs, and Kazakhs accounting for more than 90% of the total population. Due to differences in customs and living environments, there is a certain heterogeneity in the dietary structures of these different ethnic groups [13]. Moreover, Xinjiang is located in the northwest of China, which, together with the developed animal husbandry industry in the region, has resulted in a diet that favors livestock and poultry meat. The dietary survey was refined by combining the 32 common food groups published by the Chinese Nutrition Society and the dietary characteristics of Xinjiang, and the questionnaire eventually included 127 food items (e.g., aquatic products, livestock and poultry meat, fresh fruits/vegetables, root vegetables, cereal crops, soy products/legumes, nuts, etc.). With regard to the dietary influences on antibiotic exposure, the main categories included seafood, livestock, and poultry meat (beef, lamb, pork), eggs, milk, vegetables, and fruits, in conjunction with previous studies [7,14,15,16,17]. The frequency of consumption of eggs and milk was determined based on whether they were consumed ≤1–3 times/month or ≥1–3 times/week; the frequency of consumption of vegetables and fruits was determined based on whether they were eaten every day or not eaten every day; and foods such as pork and seafood were categorized according to whether they were eaten occasionally or not eaten. Furthermore, the sources of drinking water were categorized as tap water, well water, and river water.

The plant-based diet index (PDI) was also introduced in this study. We scored the PDI based on the frequency of intake. Furthermore, studies have shown that the use of non-quantitative food frequency questionnaires to assess dietary patterns is reliable and valid [18,19]. We assigned a score of 5 to the most frequent consumption of plant foods and 1 to the least frequent consumption of plant foods (positive score), and a score of 1 to the most frequent consumption of animal foods and 5 to the least frequent consumption of animal foods (reverse score). A score of 5 was assigned to the consumption of whole grains, vegetable oils, and refined grains, and 1 to the consumption of animal fats [20]. Individual scores for the 18 food groups were summed to obtain an index with a theoretical range of 18 (lowest possible score) to 90 (highest possible score) [21]. Higher PDI scores indicate more frequent consumption of plant-based foods [20,22]. In addition, we cited the unhealthy PDI (uPDI) by assigning a positive score to less healthy plant-based food (refined grains, pickled vegetables, and sugar) consumption and the opposite score to healthy plant-based and animal-based foods. In contrast, higher uPDI scores reflect a higher frequency of less healthy plant-based food intake and a lower frequency of healthy plant-based food intake [22]. Total scores were calculated for the PDI and uPDI, which were then divided into four groups based on quartiles. Group Q1 of the PDI represents those who did not prefer a plant-based diet, and group Q4 represents those who strongly preferred a plant-based diet. Group Q1 of the uPDI represents those who did not prefer an unhealthy plant-based diet, and group Q4 represents those who preferred an unhealthy plant-based diet.

### 2.5. Statistical Analysis

The frequency of detection and the concentration percentile of antibiotics were calculated based on demographic characteristics. Antibiotic concentrations below the LOD were replaced by concentrations at LOD/2 [8]. The concentrations were also divided into 3 strata: stratum 1 (<LOD), stratum 2 (low concentration), and stratum 3 (high concentration). Stratum 2 and stratum 3 were separated by the median of the detected concentrations. The distribution of concentrations, in line with previous literature, was compared using the 95th percentile concentration. The chi-square test was used for categorical data and the rank sum test (Kruskal-Wallis test) was used for continuous data that did not conform to a normal distribution. Variables that were statistically significant in the univariate analysis were included in the multivariate analysis. Binary or multiple logistic regression models were used to identify independent variables associated with detected urinary antibiotic concentration and frequency of detection. Values of *p* < 0.05 were considered statistically significant. Statistical analyses were performed using IBM SPSS (version 26.0). The bar chart was produced by R-Package GGplot2 and the software version was 4.1.4.

## 3. Results

A total of 873 subjects participated in the study and the overall detection rate of urinary antibiotics was 49.8% (Table 1). The differences in total detection rate (*p* = 0.005) and detection concentration (*p* = 0.005) among the study subjects in different age groups were statistically significant (Table 2). The four age groups had the lowest to highest antibiotic detection rates of 45.3%, 45.0%, 58.3%, and 55.9%, respectively, with the highest rate of 58.3% in the 56–65 years age group. The concentration of antibiotics detected in each age group was 290, 40, 480, and 910 ng/mL, respectively, with the highest concentration of 910 ng/mL in the 66–75 years age group, which showed an increasing trend with age from 46 to 75 years. The total antibiotic detection rate was higher in the population with low education than in the population with high education, and showed a decreasing trend with increasing education level (detection rate: 57.0%, 51.7%, 41.8%, 39.0%). The highest concentration of antibiotics was detected in those with primary education (380 ng/mL) and the lowest in those with secondary education (66 ng/mL), with a significant difference in antibiotic concentrations between the different education levels (*p* = 0.02). The per capita monthly expenditure was not significantly different between the different population subgroups in terms of total detection rate and detected concentration. We also investigated the source of drinking water, frequency of eating eggs, frequency of drinking milk and goat’s milk, frequency of eating beef, frequency of eating mutton, frequency of eating seafood, frequency of eating vegetables, and frequency of eating fruits. The above factors did not show statistically significant differences in the total detection rate and detected concentration among the different population subgroups (Table 3). There was a statistically significant difference in the detection rate (*p* = 0.04) and detected concentration (*p* = 0.007) between the two groups of subjects who consumed pork and those who did not consume pork. There was a significant difference in antibiotic total detection rate (*p* = 0.02) and concentration (*p* = 0.03) in the population grouped by the PDI, and the highest detected antibiotic concentration (2500 ng/mL) was found in the population grouped by PDI in Q2, with the concentration of antibiotics showing a decreasing trend from Q2 to Q4. There was no significant difference in antibiotic total detection rate and concentration in the population grouped by uPDI, but it is noteworthy that the population grouped by uPDI showed an increasing trend in detected antibiotic concentration from Q2 to Q4.

According to the classification of antibiotic preference, PVAs were detected at the highest rate (40.5%), followed by HAs (12.3%) and VAs (10.3%). PVAs were detected at the highest concentration (51 ng/mL), followed by HAs (3.1 ng/mL) and VAs (2.2 ng/mL). Moreover, the detection rate of PVAs was significantly different between the three groups who did not eat beef, occasionally ate beef, and ate beef every day, respectively, the two groups who ate fruit every day and did not eat fruit every day, respectively, and between people of different ages and people with different levels of education. The detection rate of VAs was significantly different between the two groups who ate eggs 1–3 times per month and those who ate eggs 1–3 times per week.

The presence of at least two antibiotics in the body was detected in 20.4% of the study subjects. Individual antibiotic detection rates ranged from 0.1% to 19.2% (Appendix A). Ofloxacin (19.2%) was the most common antibiotic, while sulfadiazine (0.1%) was the least common. In terms of antibiotic class, the detection rate ranged from 8.9% to 30.1%. Fluoroquinolones (30.1%) were higher than sulfonamides (8.9%). Individual antibiotic concentrations in participants ranged from less than the LOD to 9.02 ng/mL. None of the study participants had urinary antibiotic concentrations above 10 ng/mL. The frequency and concentration of adult-specific antibiotics varied by age, literacy, consumption of pork, vegetables, fruit, PDI, and uPDI (Appendix A, Figure 1). There were also differences in the frequency and concentration of antibiotics (Table 2 and Table 3). There was a statistically significant difference in the detection rate (*p* = 0.02) and concentration (*p* = 0.002) of thiamphenicol between the pork-eating and non-pork-eating groups, as well as a statistically significant difference in the concentration of sulfadiazine between the two groups (*p* = 0.01). Oxytetracycline and norfloxacin were specific in those who ate fruit daily. There was a significant difference in the detected concentrations of doxycycline between those who ate vegetables daily and those who did not eat vegetables daily. Differences were found in the detection rates and concentrations of tetracyclines, ciprofloxacin, ofloxacin, and chloramphenicol in the PDI group. In addition, there was a statistically significant difference in the concentrations of fluoroquinolones and sulfamethoxazole in the PDI group. In contrast, there was specificity in the uPDI group with respect to both the detection rate and the detected concentration of ofloxacin. The detection rates and concentrations of fluoroquinolones and norfloxacin were generally higher in those with low education than in those with high education. There were also significant differences in the detection rates and concentrations of fluoroquinolones and norfloxacin among different age groups.

Univariate variables that were statistically significant were included in a multifactorial analysis to explore independent correlations in the detection rate and concentrations of antibiotics in human urine, and factors influencing the occurrence of urinary antibiotics (Appendix A). Occasional pork eaters were significantly associated with thiamphenicol (*OR* = 15, 95% *CI* = 1.7–131). Daily fruit eaters were more likely to have norfloxacin detected (*OR* = 0.4, 95% *CI* = 0.2–0.7) compared to those who did not eat fruit daily, and the levels of norfloxacin were statistically significant in the low concentration group (Tier 2) and high concentration group (Tier 3) compared to the below the LOD group (Tier 1) for those who ate fruit daily. In the population with a PDI, the Q3 group was significantly associated with chloramphenicol (*OR* = 0.4, 95% *CI* = 0.2–0.9), and the Q4 group was significantly associated with fluoroquinolones (*OR* = 0.6, 95% *CI* = 0.4–0.99) and chloramphenicol (*OR* = 0.2, 95% *CI* = 0.1–0.6). In addition, the Q2 group was statistically significantly associated with high fluoroquinolone concentration levels (Tier 3) compared to those in the group below the LOD (Tier 1).

## 4. Discussion

With the aim of preventing antibiotic resistance, our research concerned the impact of antibiotics on human health. This study investigated urinary antibiotic exposure and potential determinants of antibiotic exposure in adults in Xinjiang. The detected concentrations and detection rates of 18 antibiotics were measured and compared in different adult groups. Similar to previous reports, we observed a large variation in the detection rates of different antibiotics [7,23,24]. The overall detection rate of antibiotics was 49.8% in the 873 included adults. Of these, fluoroquinolones (used as VAs or PVAs) and tetracyclines (used as VAs or PVAs) were the most common by category. In terms of individual antibiotics, the highest detection rate (30.1%) was found for ofloxacin (PVA), and the highest concentration (17 ng/mL) was detected for tetracycline (PVA).

In this study, at least two antibiotic residues were detected in 20.4% of the population, indicating widespread exposure of adults in Xinjiang to multiple environmental antibiotics. Despite the potential human health effects of exposure to antibiotics in the daily environment (diet, drinking water), data from comprehensive biomonitoring-based investigations of human antibiotic exposure remain limited. In fact, to date, human antibiotic exposure has only been reported in China and Korea [6,7,8,14,17,23,25,26,27]. Of these studies, only one examined the level of antibiotic exposure in adults [6], and none examined factors influencing antibiotic exposure in adults. Only a few articles have examined the factors influencing antibiotic exposure in children and pregnant women. Regarding exposure levels, the exposure concentrations of fluoroquinolones and ofloxacin in adults in Xinjiang were significantly lower than the exposure concentrations of fluoroquinolones and ofloxacin in adults in Shanghai determined by Wang et al. The exposure concentrations of tetracyclines were higher than the exposure concentrations in adults in Shanghai [6]. However, in another study by Zeng et al. on antibiotics in pregnant women in Shanghai, the concentration levels of fluoroquinolones, ofloxacin, and tetracyclines in adults in Xinjiang were higher than the exposure levels in pregnant women in Shanghai [7]. The concentrations of tetracyclines in adults in Xinjiang were higher than the concentrations of antibiotics in children and pregnant women in Jiangsu [8]. The reasons for these variations in antibiotic exposure may be related to the degrees of environmental and food contamination by antibiotics in different regions (e.g., major antibiotics used by humans and animals), the levels of economic development, the dietary habits and lifestyles of local people, and knowledge of antibiotic safety [28,29,30,31,32]. Specifically, as fluoroquinolones are a major factor in surface water contamination, aquatic products are more susceptible to fluoroquinolone contamination. Shanghai is closer to the sea and the aquaculture industry is more developed in Shanghai than Xinjiang; thus, the concentration of fluoroquinolones in adults in Xinjiang was lower than that in Shanghai [33,34]. Tetracyclines are the most commonly used antibiotics in animal husbandry, mainly for promoting growth and preventing infectious diseases in animals. Compared with Shanghai and Jiangsu, the livestock industry in Xinjiang is more developed, and people have a relatively high intake of livestock and poultry meat, so the detected concentrations of tetracyclines were relatively high.

In addition, we detected samples of HAs that are not commonly used as veterinary drugs (azithromycin, clarithromycin, roxithromycin, chloramphenicol). These HAs have also been detected in adults in other parts of China (3.8%, 0.5%, 3.3%, and 5.8% vs. 1.6%, 0.2%, 0.2%, and 2.3%) [6]. Notably, we detected the above HAs antibiotics in people who regularly ate pork, vegetables, and fruits, with a detection rate of up to 7.4%. Based on previous reports in the literature, it is possible that the accumulation of clinically used antibiotics constitutes a source of exposure to HAs [17,23]. However, based on the inclusion and exclusion criteria of this study, as well as the information from the questionnaires obtained face-to-face, it is known that the included subjects did not use antibiotics within one month of the study. Therefore, we can conclude that the detected HAs entered the body through the ingestion of antibiotic-contaminated food or drinking water.

Dietary intake is considered to be one of the important routes of chronic low-dose exposure to antibiotics in humans [17,25]. Consistent with the results of this study, the consumption of foods of animal origin has been reported to be associated with increased detection rates of VAs and PVAs in children [17]. In this study, we found that the detection rates of PVAs and VAs were different for participants with different frequencies of eating beef and different frequency groupings of eating eggs. This may be due to the fact that PVAs and VAs were administered during the rearing of cattle and chickens, resulting in antibiotic residues in beef and eggs. In the study in Jiangsu, China, the detection rate of veterinary antibiotics was also higher in eggs and beef [35]. Therefore, these veterinary foods are a likely source of antibiotic exposure [25]. In addition, tests of *E. coli* in pigs from large pig farms in several Chinese provinces showed very high antibiotic detection rates, especially in Henan and Hubei provinces [36]. This supports our results showing that thiamphenicol is likely to be added as feed during pig rearing. Therefore, people who eat pork regularly are more likely to have residual thiamphenicol [37]. Our study also found a significant difference in the detection rate of PVAs between those who ate fruit daily and those who did not eat fruit daily, and daily fruit consumption was associated with the detection rate of norfloxacin (used as a PVA), and showed a dose-response relationship (Appendix A). Although antibiotic studies in fruits have not been reported previously, antibiotic residues in the soil where fruits are grown and residues of antibiotics in pesticides could still be potential causes of antibiotic enrichment in fruits [38]. Consistent with previous reports examining vegetables as an influential factor, eating vegetables was not significantly associated with antibiotic exposure. However, our study also detected a significant difference in doxycycline concentrations between the groups who ate vegetables daily and those who did not (Appendix A). This is consistent with a previous study in which antibiotics were also detected in mung bean sprouts, suggesting that antibiotics tend to accumulate in vegetable-based foods [39]. The reason for this discrepancy may be due to the fact that doxycycline is used to sterilize vegetables during the growing process, which in turn leads to their absorption and enrichment, and then the antibiotic-contaminated vegetables enter the human body through the dietary route, resulting in antibiotic residues in the human body [40]. In addition, we applied two dietary pattern constructs to corroborate the relationship between dietary pattern (a plant-based diet (PDI) and an unhealthy plant-based diet (uPDI)) and antibiotics. We found lower antibiotic concentrations in people with a preference for a plant-based diet, but higher antibiotic concentrations in those with a preference for an unhealthy plant-based dietary pattern, suggesting that a non-preference for the plant-based diets and unhealthy plant-based diets are important conditions for antibiotic intake in the body. Our study found that the Q2 group, which disliked the PDI, was associated with fluoroquinolones at high concentration levels (Tier 3). Consistent with the results of this study, fluoroquinolones have been detected in meat samples [41]. In addition, previous studies have shown that a higher uPDI is associated with higher intakes of red meat, processed meat, and dairy products [42]. Consistent with previous studies, we also found a significant difference in the detected concentrations of sulfadiazine between those who ate pork and those who did not. This may be due to the fact that sulfadiazine is an important drug used for bactericidal purposes during pig feeding, and therefore residual sulfadiazine is found in pork.

This biomonitoring-based study reflected the level of antibiotic exposure among adults living in Xinjiang. None of the participants reported using any of the antibiotics analyzed within one month of the study, ruling out interference of iatrogenic antibiotics. However, there are some limitations in this study. First, antibiotic exposure in adults may be underestimated as some other common antibiotics, such as β-lactams and other types of tetracycline, were not analyzed in this study. However, these antibiotics have been detected in water environments in China and European countries in previous studies, and human exposure to environmental antibiotics may be higher [33,43]. However, β-lactams were not considered in this study due to the limited use of cephalosporin in animal husbandry and its widespread clinical use [7]. Secondly, the food frequency questionnaire may have led to recall bias. As a result, estimates of the true association between food intake and antibiotic levels in urine may be slightly inaccurate.

## 5. Conclusions

The current study showed that adults in Xinjiang are widely exposed to a variety of antibiotics. Fluoroquinolones were found to be the most important antibiotics in terms of exposure, followed by tetracyclines. Pork and fruit intake may be potential sources of antibiotic exposure. Further research is needed to investigate the health effects of antibiotics in humans.

## Figures and Tables

**Figure 1 nutrients-14-04748-f001:**
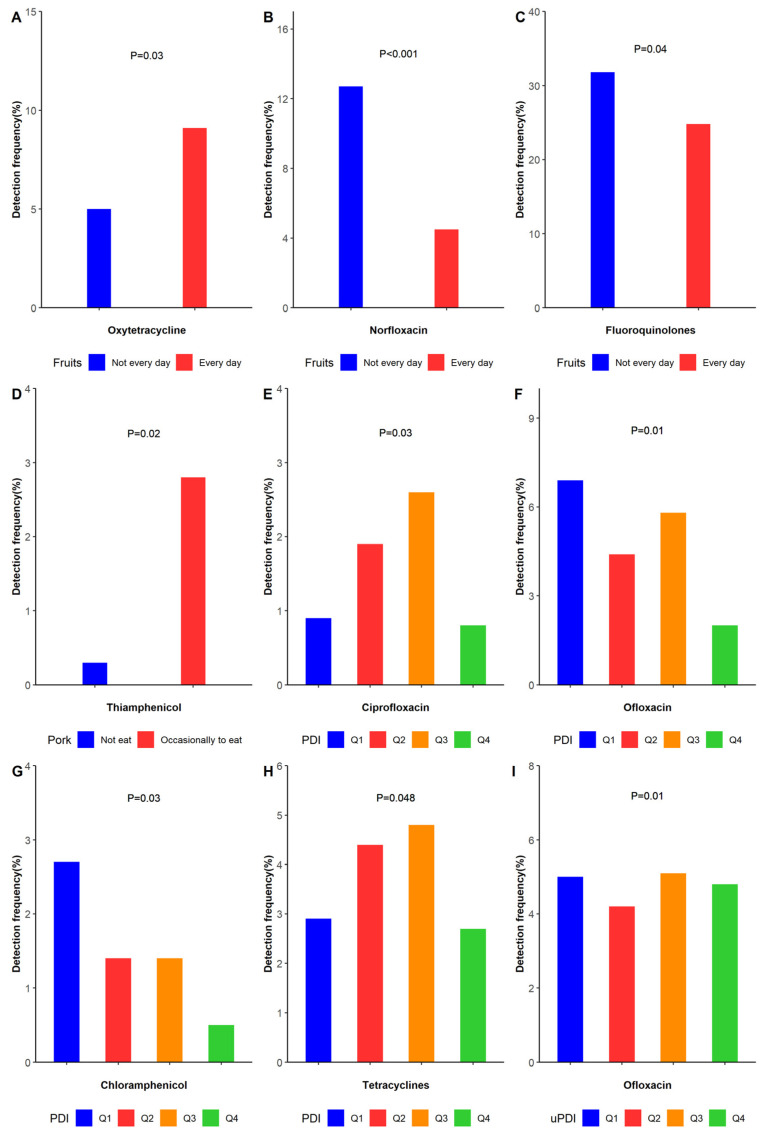
The rate of detection of antibiotics in urine was related to selected dietary characteristics. (The results in Figure 1 are based on the positive results in Appendix A). (**A**) Eating fruits every day and not every day in the detection frequency of oxytetracycline. (**B**) Eating fruits every day and not every day in the detection frequency of norfloxacin. (**C**) Eating fruits every day and not every day in the detection frequency of fluoroquinolones. (**D**) Eating pork occasionally and not eating pork in the detection frequency of thiamphenicol. (**E**) Detection frequency of four groups of PDI in ciprofloxacin. (**F**) Detection frequency of four groups of PDI in ofloxacin (**G**) Detection frequency of four groups of PDI in chloramphenicol. (**H**). Detection frequency of four groups of PDI in tetracyclines. (**I**) Detection frequency of four groups of uPDI in ofloxacin.

**Table 1 nutrients-14-04748-t001:** Detected Frequency and Concentration Distribution of Antibiotics in the Urine of Adults in Xinjiang (*n* = 873).

Antibiotic	Usage	N(%) ^a^	Concentration (ng/mL)					
			Percentiles					Maximum
			65th	75th	85th	95th	99th	
All antibiotics ^b^		435(49.8)	0.5	1.8	8	220	21,000	80,000
HAs		107(12.3)	—	—	—	3.1	470	19,000
VAs		90(10.3)	—	—	—	2.2	95	36,000
PVAs		354(40.5)	0.3	0.8	3.8	51	12,000	80,000
Tetracyclines ^c^		128(14.7)	—	—	—	17	6900	45,000
Chlortetracycline	VA	5(0.6)	—	—	—	—	—	120
Tetracycline	PVA	96(11)	—	—	—	9	390	44,000
Doxycycline	PVA	6(0.7)	—	—	—	—	—	11,000
Oxytetracycline	VA	54(6.2)	—	—	—	1.5	72	36,000
Fluoroquinolones ^c^		263(30.1)	—	0.3	0.8	7.7	550	24,000
Enrofloxacin	VA	12(1.4)	—	—	—	—	0.2	9.4
Norfloxacin	PVA	92(10.7)	—	—	—	1	51	24,000
Ciprofloxacin	PVA	53(6.1)	—	—	—	0.6	6.9	930
Ofloxacin	PVA	168(19.2)	—	—	0.2	1.6	34	2900
Macrolides ^c^		61(7)	—	—	—	2.1	470	19,000
Azithromycin	HA	33(3.8)	—	—	—	—	100	650
Clarithromycin	HA	4(0.5)	—	—	—	—	—	4600
Roxithromycin	HA	29(3.3)	—	—	—	—	82	19,000
Sulfonamides ^c^		78(8.9)	—	—	—	0.3	2.8	80,000
Sulfamethazine	PVA	20(2.3)	—	—	—	—	0.3	12
Sulfadiazine	PVA	1(0.1)	—	—	—	—	—	11
Sulfamethoxazole	PVA	18(2.1)	—	—	—	—	1.3	78,000
Trimethoprim	PVA	58(6.6)	—	—	—	0.1	2.1	2000
Phenicols ^c^		83(9.5)	—	—	—	0.1	1.8	62
Chloramphenicol	HA	51(5.8)	—	—	—	0.04	0.7	60
Florfenicol	VA	33(3.8)	—	—	—	—	0.6	62
Thiamphenicol	PVA	5(0.6)	—	—	—	—	—	11

HAs: Human Antibiotics; VAs: Veterinary Antibiotics; PVAs: Preferred Veterinary Antibiotics. ^a^ Positive detection (detection frequency, %). ^b^ Positive detection of one or more studied antibiotics. ^c^ Positive detection of one or more antibiotics in corresponding category. — Urinary concentration of antibiotics is below the limit of detection (LOD).

**Table 2 nutrients-14-04748-t002:** Overall Detection Rate and Concentrations of Urinary Antibiotics in Relation to Selected Characteristics.

Variable	Overall ^a,b^	Antibiotic Concentration (ng/mL) ^c,d^	Has ^a,b^	Vas ^a,b^	PVAs ^a,b^
Sex					
Male	210(49.2)	250	54(12.6)	45(10.5)	170(39.8)
Female	225(50.4)	87	53(11.9)	45(10.1)	184(41.3)
Age (years) *^^‡^					
35–45	107(45.3)	290	39(16.5)	20(8.5)	79(33.5)
46–55	138(45.0)	40	30(9.8)	32(10.4)	114(37.1)
56–65	133(58.3)	480	25(11.0)	26(11.4)	115(50.4)
66–75	57(55.9)	910	13(12.7)	12(11.8)	46(45.1)
Categories of BMI					
Normal weight	235(52.2)	290	53(11.8)	51(11.3)	189(42.0)
Obesity	200(47.3)	110	54(12.8)	39(9.2)	165(39.0)
Education *^^‡^					
<Primary	126(57.0)	250	29(13.1)	25(11.3)	105(47.5)
Primary	199(51.7)	380	57(14.8)	36(9.4)	158(41.0)
Secondary	87(41.8)	66	16(7.7)	24(11.5)	72(34.6)
≥High school	23(39.0)	180	5(8.5)	5(8.5)	19(32.2)
Monthly expenditure per capita (RMB)					
≤240	117(53.7)	500	27(12.4)	22(10.1)	95(43.6)
240–333.33	82(49.7)	180	17(10.3)	20(12.1)	70(42.4)
>333.33	230(48.0)	150	61(12.7)	46(9.6)	186(38.8)

^a^ Number of positive detections (detection frequency, %). ^b^ Chi-square test. ^c^ The 95th percentile urinary concentration (ng/mL). ^d^ Non-parametric comparisons (Mann–Whitney test, Kruskal–Wallis test). * *p* < 0.05 vs. detection rate. ^ *p* < 0.05 vs. antibiotic concentration. ^‡^ *p* < 0.05 vs. PVAs.

**Table 3 nutrients-14-04748-t003:** Overall Detection Rate and Concentrations of Urinary Antibiotics in Relation to Selected Characteristics.

Variable	Overall ^a,b^	Antibiotic Concentration (ng/mL) ^c,d^	Has ^a,b^	Vas ^a,b^	PVAs ^a,b^
Source of drinking water					
Tap water	308(48.8)	100	71(11.3)	65(10.3)	249(39.5)
Well and river water	125(53.0)	320	35(14.8)	25(10.6)	103(43.6)
Eggs and products ^†^					
≤1–3 times/month	206(49.5)	290	50(12.0)	34(8.2)	171(41.1)
≥1–3 times/week	221(50.3)	93	54(12.3)	54(12.3)	176(40.1)
Cow and Goat Milk					
≤1–3 times/month	269(51.0)	230	70(13.3)	54(10.2)	216(41.0)
≥1–3 times/week	153(48.1)	120	32(10.1)	33(10.4)	129(40.6)
Pork *^					
Not eaten	367(50.8)	240	91(12.6)	72(10.0)	299(41.4)
Occasionally eaten	44(40.4)	30	7(6.4)	12(11.0)	36(33.0)
Beef ^‡^					
Not eat	117(50.0)	420	26(11.1)	33(14.1)	88(37.6)
Occasionally eaten	233(48.3)	110	67(13.9)	41(8.5)	190(39.4)
Eaten every day	73(55.7)	580	11(8.4)	13(9.9)	65(50.4)
Mutton					
Not eat	84(50.3)	620	21(12.6)	20(12.0)	72(43.1)
Occasionally eaten	170(48.2)	80	41(11.6)	34(9.6)	136(38.5)
Eaten every day	170(51.7)	280	42(12.8)	34(10.3)	137(41.6)
Seafood					
Not eaten	254(50.6)	490	65(12.9)	51(10.2)	206(41.0)
Occasionally eaten	161(48.6)	74	35(10.6)	36(10.9)	133(40.2)
Vegetables					
Not every day	10(40.0)	180	4(16.0)	2(8.0)	6(24.0)
Every day	415(50.1)	220	101(12.2)	86(10.4)	339(40.9)
Fruits ^‡^					
Not every day	313(50.8)	180	75(12.2)	57(9.3)	262(42.5)
Every day	115(47.5)	270	31(12.8)	31(12.8)	85(35.1)
PDI *^					
Q1	125(50.2)	90	34(13.7)	20(8.0)	98(39.4)
Q2	98(49.5)	2500	23(11.6)	24(12.1)	80(40.4)
Q3	143(56.3)	290	37(14.6)	29(11.4)	118(46.5)
Q4	65(40.6)	63	12(7.5)	16(10.0)	54(33.8)
uPDI					
Q1	134(48.6)	470	35(12.7)	30(10.9)	104(37.7)
Q2	113(48.5)	80	29(12.4)	26(11.2)	92(39.5)
Q3	88(52.4)	450	22(13.1)	19(11.3)	71(42.3)
Q4	96(52.2)	740	20(10.9)	14(7.6)	83(45.1)

^a^ Number of positive detections (detection frequency, %). ^b^ Chi-square test or Fisher test. ^c^ The 95th percentile urinary concentration (ng/mL). ^d^ Non-parametric comparisons (Mann–Whitney test, Kruskal–Wallis test). * *p* < 0.05 vs. detection rate. ^ *p* < 0.05 vs. antibiotic concentration. ^†^ *p* < 0.05 vs. VAs. ^‡^ *p* < 0.05 vs. PVAs.

## Data Availability

Data available on request due to restrictions eg privacy or ethical.

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
