# Peer review of "Urinary Antibiotics and Dietary Determinants in Adults in Xinjiang, West China"

_nutrients, 2022, doi:10.3390/nu14224748_

Round 1

Reviewer 1 Report

In such a time, when bacteria are becoming more and more resistant to wide-spectrum antibiotics, the problem of the antibiotics intake in daily meals is noteworthy and should be emphasized. I find the paper very significant for current knowledge. 

I want to point out some remarks.

Please rephrase the sentence in lines 12-14. The sentence may lead to a misunderstanding that the antibiotics were detected in all samples (873). Additionally, provide the information that the antibiotics were detected in almost half of the collected samples. The results in the abstract are too detailed. I propose to shorten lines 21-25 to one sentence, as it may encourage the reader to read the full version.

The authors tested 873 patients from Xinjiang but did not provide information about the region's population. Please add it to calculate the percentage of tested people in this region, and it may help to visualize the scale of the problem.

Explain the difference between PVAs and VAs in the introduction.

Add to the introduction a table or a graph presenting which antibiotic was classified into the groups: PVA, VA, HA.

Line 147 Explain what the "less healthy plants are."

Line 237 "literacy"? Or you meant "education"?

The results are clearly described, but the tables are too extensive and detailed. Commonly used symbols may replace the p-value *, **, *** for values statistically significant, and there is no need to provide the p-value for insignificant values. Moreover, there is no need to give the total 'n' when you provide the number of patients in each group with the percentage.

In the discussion, the Authors should be more focused on explaining the sources of the high antibiotics intake in each described group of patients than on presenting the results. The authors should extend the discussion. In Lines 319-320, the authors mentioned that the accumulation of clinically used antibiotics might bias the results. Could you provide the information on how the antibiotics are accumulated? Could you propose how to minimize the bias? I do not fully agree with the statement in line 323. To be sure that the detected Has come only from food or drinking water, the authors should provide data that the Has do not accumulate for more than 1 month or extend the period of dechallenge in the exclusions.

Finally, the authors should compare the data with current articles. Almost half of the cited references (18/40) were published more than five years ago. 

Author Response

   We sincerely thank the editor and all reviewers for their valuable comments on our manuscript entitled "Urinary Antibiotics and Dietary Determinants in Adults in Xinjiang, West China ". Those comments are highly insightful and constructive, and enabled us to greatly improve the quality of our manuscript. We have addressed the comments very carefully. All the changes made in the revised manuscript are highlighted in yellow. The revisions in the manuscript and the responses to the reviewers' comments are listed below.

Point 1: In such a time, when bacteria are becoming more and more resistant to wide-spectrum antibiotics, the problem of the antibiotics intake in daily meals is noteworthy and should be emphasized. I find the paper very significant for current knowledge. 

I want to point out some remarks.

Please rephrase the sentence in lines 12-14. The sentence may lead to a misunderstanding that the antibiotics were detected in all samples (873). Additionally, provide the information that the antibiotics were detected in almost half of the collected samples.

Response 1: We feel very grateful for your critical comments on our analysis results. To address your concerns, we will change this sentence to “In this study, 18 antibiotics, including four human antibiotics (HAs), four veterinary antibiotics (VAs) and 10 preferred veterinary antibiotics (PVAs) were detected in the urine of approximately half of the 873 adults in Xinjiang including Han Chinese (24.6%), Hui (25.1%), Uighur (24.6%) and Kazakh (25.7%).” All the changes made in the revised manuscript are highlighted in yellow. (pages 12-15)

The specific data presentation is shown in Table 1 (marked yellow), and the results of this data are also mentioned in lines 17-18 of the abstract "The detection percent of antibiotics in the urine of adults in Xinjiang ranged from 0.1% to 30.1% with a total detection percent of all antibiotics of 49.8%." (marked yellow)

Point 2: The results in the abstract are too detailed. I propose to shorten lines 21-25 to one sentence, as it may encourage the reader to read the full version.

Response 2: Thank you for this suggestion. The result in lines 21-25 has been changed to a sentence “Adults who regularly ate pork, daily fruit consumption, and did not prefer a plant-based diet were associated with thiamphenicol, norfloxacin, and fluoroquinolones, respectively.” All the changes made in the revised manuscript are highlighted in yellow. (pages 21-23)

Point 3: The authors tested 873 patients from Xinjiang but did not provide information about the region's population. Please add it to calculate the percentage of tested people in this region, and it may help to visualize the scale of the problem.

Response 3: We are grateful for the proposal. Since 2018, our group has conducted the National Key Research and Development Program “The Xinjiang Multiethnic Cohort (XMC) study” in Xinjiang. A sub-sample was recruited from the Yili region during the baseline survey in 2019. The recruited indi-viduals were the residents of three townships in Huocheng County of Ili (including Lan-gan Township, Sarbulak Township and Luchaogou Township), who were randomly re-cruited according to the local main ethnic groups. Inclusion criteria were as follows: adults aged 35–75 years; residents who had lived locally for three years or more; not in an acute state of illness; and not having used antibiotics within one month (to minimize the effect of clinically used antibiotics). After excluding participants with severe liver or kid-ney disease or mental illness, and those with incomplete questionnaires or whose urine or blood samples were not collected, a total of 873 adults with complete data was included in this study. We do not have specific data on the total population of the area. But for the three townships we recruited a total of 7,395 people entered our study. Therefore, the percentage of tested persons in the area is 11.81%.

Point 4: Explain the difference between PVAs and VAs in the introduction.

Response 4: Thank you for this suggestion. The meanings of PVAs and VAS are explained in the notes to Table 1 (marked in yellow). To address your concern, the difference between the two is that VAS is an animal-specific antibiotic, while PVAs are antibiotics that can be used by both humans and animals. However, animals use them mostly, followed by humans.

Point 5: Add to the introduction a table or a graph presenting which antibiotic was classified into the groups: PVA, VA, HA.

Response 5: We are grateful for the suggestion. Exactly which antibiotic was classified in which group is presented in Table 1. (marked yellow)

Point 6: Line 147 Explain what the "less healthy plants are."

Response 6: We sincerely appreciate this valuable comment. We added that the less healthy plant foods are refined grains, pickled vegetables and sugar. (marked yellow). (pages 143).

Point 7: Line 237 "literacy"? Or you meant "education"?

Response 7: We appreciate your comments. Yes, "literacy" refers to "education".

Point 8: The results are clearly described, but the tables are too extensive and detailed. Commonly used symbols may replace the p-value *, **, *** for values statistically significant, and there is no need to provide the p-value for insignificant values. Moreover, there is no need to give the total 'n' when you provide the number of patients in each group with the percentage.

Response 8: We are very grateful for your suggestions. Tables 2 and 3 in the text, Supplementary Tables S1-S4 have been revised in accordance with your suggestions.

Point 9: In the discussion, the Authors should be more focused on explaining the sources of the high antibiotics intake in each described group of patients than on presenting the results. The authors should extend the discussion.

Response 9: We appreciate your criticism of our discussion. We have therefore removed the p-values and a part of the description of the results from the discussion. The discussion has been expanded and the corresponding references have been added. The added parts have been marked in yellow. Thus, the added sentences are in lines 326-327, 328-331, and 342-344, respectively:

Lines 326-327: "In the study in Jiangsu, China, the detection rate of veterinary antibiotics was also higher in eggs and beef [35]."

Lines 328-331: " In addition, tests of E. coli in pigs from large pig farms in several Chinese provinces showed very high antibiotic detection rates, especially in Henan and Hubei provinces [36]. This supports our results that thiamphenicol is likely to be added as feed during pig rearing. "

Lines 342-344: " This is consistent with a previous study in which antibiotics were also detected in mung bean sprouts, suggesting that antibiotics tend to accumulate in vegetable-based foods [39]."

Point 10: In Lines 319-320, the authors mentioned that the accumulation of clinically used antibiotics might bias the results. Could you provide the information on how the antibiotics are accumulated? Could you propose how to minimize the bias? I do not fully agree with the statement in line 323. To be sure that the detected Has come only from food or drinking water, the authors should provide data that the Has do not accumulate for more than 1 month or extend the period of dechallenge in the exclusions.

Response 10: We appreciate your valuable questions. This is because HAs stands for human antibiotics, which are drugs taken by people. People who take drugs (HAs) cause the accumulation of antibiotics. This study was to investigate the accumulation of environmental antibiotics, i.e., antibiotics in people's daily diet and drinking water. To minimize bias, so we excluded people who took antibiotics within a month when we recruited trained medical personnel to conduct face-to-face questionnaires with the respondents. In lines 60-61, it has been marked yellow.

Point 11: Finally, the authors should compare the data with current articles. Almost half of the cited references (18/40) were published more than five years ago. 

Response 11: Thank you for your valuable advice. We have made updates to the references in 18-43, five years ago, except for a few of the more representative articles in the field (23-27). The specific references that have been changed are marked in yellow in the article.

We are very grateful for your and reviewers' warm work earnestly. We highly appreciate your valuable and constructive comments and hope that the revised version of the manuscript will meet the requirements of the journal.

We shall look forward to hearing from you at your earliest convenience.

Sincerely yours,

Professor Jianghong Dai, PhD

Department of Epidemiology and Biostatistics 

School of Public Health

Xinjiang Medical University

Urumqi, 830001, China

Reviewer 2 Report

The information presented is of general interest, but without additional data is of limited value and the results not surprising. The following information would be of much more value to the literature:

1-Is there any evidence between those who had antibiotic exposure and incidence of any infection? And if so, were any of those infections due to antibiotic-resistant bacteria?

2-Is there any evidence of adverse effects, or any differences in the health of individuals who had antibiotic exposure and those without antibiotic exposure.

This information should be available in the authors' data banks; without this additional information, this report is of limited value. And as a minimum, there should be some discussion regarding these important related issues.

Author Response

   We sincerely thank the editor and all reviewers for their valuable comments on our manuscript entitled "Urinary Antibiotics and Dietary Determinants in Adults in Xinjiang, West China ". Those comments are highly insightful and constructive, and enabled us to greatly improve the quality of our manuscript. We have addressed the comments very carefully. All the changes made in the revised manuscript are highlighted in yellow. The revisions in the manuscript and the responses to the reviewers' comments are listed below.

Point 1: The information presented is of general interest, but without additional data is of limited value and the results not surprising. The following information would be of much more value to the literature:

there any evidence between those who had antibiotic exposure and incidence of any infection? And if so, were any of those infections due to antibiotic-resistant bacteria?

Response 1: We appreciate the advice you have provided us. With the gradual increase in global consumption of antibiotics, China accounts for a high proportion of global antibiotic use. In addition to clinical use, the vast majority of antibiotics are used to treat and prevent bacterial infections in animal husbandry and aquaculture. As a result, antibiotics may re-enter the human body through unintentional ingestion of antibiotic-contaminated food or drinking water, resulting in long-term persistent low-dose antibiotic exposure, with potential adverse effects on human health. Xinjiang is located in northwest China, and has a unique ethnic structure and a well-developed livestock industry; thus, there is a higher preference for livestock and poultry meat in this region. As a result, residents of the Xinjiang region may be at greater risk of antibiotic-containing foods of animal origin.

    Exposure to antibiotics can occur through food consumption and may be associated with sociodemographic factors. It is unclear which factors are most associated with antibiotic concentrations. We sought to investigate the antibiotic exposure profile in adults in Xinjiang and to explore the potential determinants of urinary antibiotics in adults using biomonitoring data of urinary antibiotics.

    Our study primarily reflects the level of antibiotic exposure (clinical use and environmental antibiotics) and potential risk in the population and has not yet addressed the relationship between infection and antibiotics.

Point 2: there any evidence of adverse effects, or any differences in the health of individuals who had antibiotic exposure and those without antibiotic exposure.

This information should be available in the authors' data banks; without this additional information, this report is of limited value. And as a minimum, there should be some discussion regarding these important related issues.

Response 2: Thank you for your valuable suggestions. In our study, we investigated the level of human exposure to antibiotics during daily diet and water intake and examined the dietary determinants of human exposure to antibiotics. It is to prevent antibiotic resistance, and therefore we are highly concerned about the impact of antibiotics on human health, that we conducted this study. But so far, we haven't addressed the relationship between infection and antibiotics. We really don't have these data that you mentioned. We will consider further analysis later when we have data. To address your concern, we have included this study objective in the discussion, in lines 268-269: “In order to prevent antibiotic resistance, we are highly concerned about the impact of antibiotics on human health.”

We are very grateful for your and reviewers' warm work earnestly. We highly appreciate your valuable and constructive comments and hope that the revised version of the manuscript will meet the requirements of the journal.

We shall look forward to hearing from you at your earliest convenience.

Sincerely yours,

Professor Jianghong Dai, PhD

Department of Epidemiology and Biostatistics 

School of Public Health

Xinjiang Medical University

Urumqi, 830001, China

Reviewer 3 Report

This is an interesting report with regard to antibiotic exposure from diet. There are not very many, and this is welcomed. The study is well performed and the manuscript is well written. I have no amendments suggestions .

Minor points

Line 42: A number of studies have investigated; change to: A number of studies has investigated

L 67: A total of 873 adults with complete data were included; change to: A total of 873 adults with complete data was included

L 86+87+88: …and body fat percentage and muscle mass were measured using the bioelectrical impedance method. Body fat percentage and muscle mass were measured by bioelectrical impedance (instrument model: TANITA DC-430MA, Japan); change to: This sentence is a doublet; remove one of them

Author Response

   We sincerely thank the editor and all reviewers for their valuable comments on our manuscript entitled "Urinary Antibiotics and Dietary Determinants in Adults in Xinjiang, West China ". Those comments are highly insightful and constructive, and enabled us to greatly improve the quality of our manuscript. We have addressed the comments very carefully. All the changes made in the revised manuscript are highlighted in yellow. The revisions in the manuscript and the responses to the reviewers' comments are listed below.

Point 1: This is an interesting report with regard to antibiotic exposure from diet. There are not very many, and this is welcomed. The study is well performed and the manuscript is well written. I have no amendments suggestions.

Minor points

Line 42: A number of studies have investigated; change to: A number of studies has investigated

L 67: A total of 873 adults with complete data were included; change to: A total of 873 adults with complete data was included

L 86+87+88: …and body fat percentage and muscle mass were measured using the bioelectrical impedance method. Body fat percentage and muscle mass were measured by bioelectrical impedance (instrument model: TANITA DC-430MA, Japan); change to: This sentence is a doublet; remove one of them

Response 1: Thank you for finding our mistakes and suggestions for the article. Changes have been made in accordance with the suggestions you gave, and the corrected parts have been marked yellow.

We are very grateful for your and reviewers' warm work earnestly. We highly appreciate your valuable and constructive comments and hope that the revised version of the manuscript will meet the requirements of the journal.

We shall look forward to hearing from you at your earliest convenience.

Sincerely yours,

Professor Jianghong Dai, PhD

Department of Epidemiology and Biostatistics 

School of Public Health

Xinjiang Medical University

Urumqi, 830001, China
